# STAT3 Dysregulation in Mature T and NK Cell Lymphomas

**DOI:** 10.3390/cancers11111711

**Published:** 2019-11-02

**Authors:** Angelina Seffens, Alberto Herrera, Cosmin Tegla, Terkild B. Buus, Kenneth B. Hymes, Niels Ødum, Larisa J. Geskin, Sergei B. Koralov

**Affiliations:** 1Department of Pathology, New York University School of Medicine, New York, NY 10016, USA; ams2494@cumc.columbia.edu (A.S.); alberto.herrera@nyumc.org (A.H.); cosmin.tegla@nyumc.org (C.T.); terkild.buus@sund.ku.dk (T.B.B.); 2Vagelos College of Physicians and Surgeons, Columbia University, New York, NY 10032, USA; 3LEO Foundation Skin Immunology Research Center, Department of Immunology and Microbiology, University of Copenhagen, DK2200 Copenhagen, Denmark; ndum@sund.ku.dk; 4Division of Hematology/Oncology, New York University School of Medicine, New York, NY 10016, USA; kenneth.hymes@nyulangone.org; 5Department of Dermatology, Columbia University, New York, NY 10032, USA; ljg2145@cumc.columbia.edu

**Keywords:** STAT3, T cell lymphoma, lymphomagenesis, JAK/STAT

## Abstract

T cell lymphomas comprise a distinct class of non-Hodgkin’s lymphomas, which include mature T and natural killer (NK) cell neoplasms. While each malignancy within this group is characterized by unique clinicopathologic features, dysregulation in the Janus tyrosine family of kinases/Signal transducer and activator of transcription (JAK/STAT) signaling pathway, specifically aberrant STAT3 activation, is a common feature among these lymphomas. The mechanisms driving dysregulation vary among T cell lymphoma subtypes and include activating mutations in upstream kinases or STAT3 itself, formation of oncogenic kinases which drive STAT3 activation, loss of negative regulators of STAT3, and the induction of a pro-tumorigenic inflammatory microenvironment. Constitutive STAT3 activation has been associated with the expression of targets able to increase pro-survival signals and provide malignant fitness. Patients with dysregulated STAT3 signaling tend to have inferior clinical outcomes, which underscores the importance of STAT3 signaling in malignant progression. Targeting of STAT3 has shown promising results in pre-clinical studies in T cell lymphoma lines, ex-vivo primary malignant patient cells, and in mouse models of disease. However, targeting this pleotropic pathway in patients has proven difficult. Here we review the recent contributions to our understanding of the role of STAT3 in T cell lymphomagenesis, mechanisms driving STAT3 activation in T cell lymphomas, and current efforts at targeting STAT3 signaling in T cell malignancies.

## 1. Introduction

T and Natural killer (NK) cell lymphomas are a heterogeneous group of malignancies that account for 10%–15% of non-Hodgkin’s lymphomas and are often associated with poor clinical outcomes [1]. They arise either in lymphoid tissues or in extranodal sites, most commonly the gastrointestinal tract, followed by the skin [2]. Pathogenic events contributing to T cell oncogenesis include deregulation of T cell receptor (TCR) associated kinases and phosphatases, Ras and Rho GTPases, costimulatory and coinhibitory proteins, epigenetic modifiers, and components of cellular metabolism pathways. Another important contribution to T cell lymphomagenesis is dysregulation of Janus tyrosine family of kinases/Signal transducer and activator of transcription (JAK/STAT) signaling, particularly dysregulation of STAT3 [3].

Signal transducer and activator of transcription (STAT) proteins are latent transcription factors that reside in the cytoplasm and are activated by receptor-associated Janus tyrosine family of kinases (JAKs) in response to cytokine signaling [4,5]. STAT monomers are phosphorylated by receptor associated kinases and dimerize, a process mediated by reciprocal interactions between the Src homology 2 (SH2) domains and phosphotyrosine residues of the STAT proteins. The STAT dimers are actively transported through the nuclear pores using importin α/β and RanGDP complexes [6]. Once in the nucleus, STAT dimers bind DNA sequences known as gamma-activation sites (GAS) in the promoters of STAT-regulated genes and initiate transcription of target genes. Normal activation of STAT proteins is both rapid and transient, with nuclear STAT proteins ultimately dephosphorylated by nuclear phosphatases and transported back to the cytoplasm [7]. Subsequent work has shown that STATs are able to exist in the cytoplasm as stable homodimers, and that they shuttle between the nucleus and cytoplasm independently of phosphorylation [8,9,10].

In T cell biology, STAT proteins function downstream of cytokine signaling to help direct the differentiation of naive CD4+ T cells into T-helper (Th) subtypes and shape the adaptive immune response [11]. However, aberrant JAK/STAT activation has been implicated in the pathogenesis and phenotype of several hematological malignancies, as reviewed previously [12]. Activating events upstream of STAT1 and STAT5 have been identified in acute lymphoblastic leukemias, while activation of STAT3 and STAT5 is seen more frequently in diverse mature T cell malignancies [13,14,15]. In this article, we review the role and proposed mechanisms of dysregulated STAT3 signaling in T cell lymphomagenesis, relevant transcriptional targets that contribute to malignant evolution, infiltration, and survival, and the therapeutic utility of targeting aberrant STAT3 signaling.

## 2. Overview of STAT3 in T Cell Lymphomas

STAT3 is an important transcription factor in peripheral T cell differentiation. It mediates signal transduction through cytokine receptors including IL-6R, IL-10, and IL-21. In CD4+ T cells, STAT3 expression is critical for RORα and RORγt driven Th17 commitment, initiated by TGF-β and IL-6 signaling and maintained by IL-6, IL-21, and IL-23 [16,17,18]. STAT3 also mediates the expression of IL-17 by mucosal associated invariant T (MAIT) cells and is critical for the maintenance of MAIT and invariant natural killer T (iNKT) cells in peripheral blood [19]. In CD8+ T cells, STAT3 plays a role in memory T cell formation and in downregulating cytotoxicity-associated transcripts [20].

STAT3 enhances pro-survival signaling essential for T cell expansion [21], but when this pathway is not regulated correctly, STAT3 can contribute to T cell lymphomagenesis [3]. In Table 1, we outline T cell lymphomas with aberrant STAT3 signaling and highlight the relevant literature demonstrating that this pathway is frequently dysregulated in these malignancies and as such represents a promising therapeutic target.

In a cohort of 169 patient-derived tumor samples from cases of peripheral T cell lymphoma (PTCL), immunohistochemistry on tumor tissue microarrays demonstrated positive phospho-(p)STAT3 staining in 38% of samples. Among the different subtypes of PTCL, anaplastic lymphoma kinase positive anaplastic large cell lymphoma (ALK+ ALCL) tumors were 93% positive, ALK− ALCL tumors were 57% positive, angioimmunoblastic T cell lymphoma (AITL) were 29% positive, and peripheral T cell lymphoma-not otherwise specified (PTCL-NOS) were 27% positive for pSTAT3 [22,23]. In ALK+ ALCL, the most frequently implicated chromosomal aberration involves a translocation between chromosomes 5q35 and 2p23, which leads to the fusion of the nucleophosmin (*NPM*) gene to the anaplastic lymphoma kinase *ALK* gene [24]. STAT3 is a target of the NPM-ALK fusion protein, and constitutive STAT3 activation is a frequent feature in ALK+ ALCL patient tumors [25,26]. While ALK− ALCL tumors do not have the NPMALK fusion kinase, pSTAT3 positivity is still present in a fraction of patients, highlighting the importance of other mechanisms that drive constitutive STAT3 activation [22]. Activated STAT3 has also been implicated in most cases of breast implant associated-ALCL (BIA-ALCL), which was a new provisional entity in the 2017 World Health Organization classification [27]. It is a distinct pathological entity from ALK− ALCL arising from the capsule and/or effusion around silicone or saline filled breast prostheses [28].

In a cohort of 116 adult T cell lymphoma/leukemia (ATLL) patients, 43% demonstrated pSTAT3 positivity, which was associated with significantly better overall survival and progression free survival [29]. In ATLL, certain recurrent genetic alterations are associated with different clinical forms of disease. While STAT3 mutations are associated with more indolent disease, mutations in TP53 and IRF4 are associated with more aggressive forms of ATLL [27].

STAT3 mutations are also common in T cell large granular lymphocytic leukemia (T-LGLL), predominantly in cases with malignant CD8+ T cell involvement [30,31]. The presence of mutations in the SH2 domain correlates with shorter time-to-treatment failure and with comorbidities such as autoimmune hemolytic anemia and rheumatoid arthritis [32].

Among the more aggressive lymphomas arising from NK or γδ T cells, NK/T cell lymphoma, nasal type (NKTCL), hepatosplenic T cell lymphoma (HS-PTCL), and primary cutaneous and peripheral γδ T cell lymphomas (PC- γδ-PTCL and γδ-PTCL), STAT3 mutations have been reported in a small fraction of cases, including activating mutations in the SH2 domain [33,34,35,36]. In NKTCL cell lines, STAT3 is constitutively phosphorylated and promotes growth and cell survival [36]. The cell lines were responsive to treatment with a JAK inhibitor, which highlights both the importance of dysregulated STAT3 in this aggressive lymphoma and suggests a therapeutic benefit to targeting the JAK/STAT pathway [36].

In a study of 69 tumors from enteropathy-associated T cell lymphoma (EATL), an aggressive malignancy and the most common neoplastic complication of Celiac disease, the JAK/STAT pathway was the most frequently mutated signaling pathway. Activating mutations in STAT3 were identified in 16% of these tumors [37]. As in other aggressive mature T cell lymphomas, JAK/STAT signaling dysregulation is a key component of EATL oncogenesis.

Cutaneous T cell lymphoma (CTCL) is a type of mature T cell lymphoma characterized by the accumulation of malignant T cells in the skin. Mycosis fungoides (MF) is an indolent malignancy, especially in its early stages, but it can progress rapidly with extracutaneous dissemination of malignant T cells. Sézary syndrome (SS) is a leukemic variant of CTCL that is characterized by a clonal population of circulating CD4+ malignant lymphocytes known as Sézary cells, which are found in the skin, lymph nodes, and peripheral blood. SS is aggressive, with a survival time of 2–4 years following diagnosis [38,39,40]. Early stages of MF are characterized by expansion of T cells with a predominantly Th1 phenotype, while later stages of disease have STAT5 driven Th2-like and STAT3 driven Th17-like phenotypes [41]. MF cell lines often display constitutively activated STAT3 and are susceptible to JAK kinase inhibition [42]. In cutaneous lesions from MF, pSTAT3 is present in dermal infiltrates from tumor stage MF. In patch/plaque stage disease, pSTAT3 is seen sporadically, and it is absent in epidermal Pautrier microabscesses of early stage MF, suggesting that pSTAT3 may drive disease progression [43]. Western blot analysis of isolated circulating Sézary cells from patients and of a patient-derived SS cell line revealed a higher pSTAT3 signal when compared to healthy control peripheral blood mononuclear cells (PBMCs) [44]. However, in primary Sézary cells from SS patients, STAT3 is not constitutively active in the absence of exogenous cytokines, unlike what has been observed in the SS patient derived cell line Se-Ax [44,45]. This points to the role of cell extrinsic factors in maintaining constitutive STAT3 signaling, including the tumor microenvironment. 

Taken together, these studies show that dysregulation of the JAK/STAT signaling pathway is a common feature among mature T cell lymphomas. There are other members of the JAK/STAT family that are dysregulated in T and NK cell lymphomas, especially STAT5B in extranodal NK/T cell lymphoma, hepatosplenic T cell lymphoma, and T-LGL [27,34]. However, in this review we focus on the role of aberrant STAT3 signaling. In patients carrying these mutations, therapeutic strategies targeting different components of the pathway or the tumor microenvironment that drives its aberrant activation may be useful.

## 3. Mechanism of Aberrant STAT3 Activation

As discussed above, dysregulated STAT3 signaling plays a role in the development and progression of many malignancies. Constitutive STAT3 phosphorylation is a common feature in many lymphomas. The mechanisms of STAT3 dysregulation vary among different types of T and NK cell lymphomas, and they include activating mutations in JAK1 and STAT3, dysregulated cytokine signaling, upregulation positive regulators, and a loss of negative regulators of STAT3. Figure 1 shows the different mechanisms of STAT3 dysregulation as well as relevant downstream target genes.

### 3.1. Activating Mutations in JAK/STAT

Activating mutations of STAT3 tend to fall in the SH2 domain and lead to dimerization and nuclear localization independent of cytokine signaling. In T-LGL, next generation sequencing revealed mutations in the SH2 domain in 31 out of 77 (40%) patients [30]. These mutations were either missense mutations or insertion mutations, but all resulted in increased activation of STAT3, as demonstrated by increased nuclear pSTAT3 in bone marrow biopsies and in Western blots. These patients also had an increased incidence of neutropenia and rheumatoid arthritis, two common comorbidities in T-LGL [30]. Another study of 120 patients with T-LGL found that 27% had mutations in the SH2 domain of STAT3. However, treatment of cells from patients with a STAT3 inhibitor, STA-21, led to increased apoptosis of malignant cells irrespective of STAT3 mutation status with little impact on healthy cells [32]. This suggests that there are other mechanisms controlling STAT3 dysregulation in T-LGL besides STAT3 activating mutations. Activating mutations outside of the SH2 domain have been identified in T-LGL in the DNA-binding and coiled-coil domains, although this still does not account for the full spectrum of STAT3 activation in this malignancy [65]. 

Activating mutations in the SH2 domain of STAT3 are seen in other T cell lymphomas as well. A combination of RNA sequencing, whole exome sequencing, and Sanger sequencing on tumor samples from patients with NKTCL revealed that 5.9% had activating STAT3 mutations, while 8.3% of tumors from γδ PTCL patients harbored these mutations. Analysis of NKTCL cell lines revealed that three out of six lines had activating mutations in the SH2 domain of STAT3, which leads to high levels of pSTAT3. Treatment of an NK cell line transformed with mutant STAT3 with a JAK 1/2 inhibitor, AZD1480, resulted in dose-dependent growth inhibition [34]. Additionally, a study of 37 samples of extranodal NKTCL revealed activating mutations in the SH2 domain of STAT3 in seven cases. These samples also had strong pSTAT3 expression, indicating that the mutation in the SH2 domain was able to drive JAK/STAT signaling [67]. Treatment of two NK/T cell lymphoma cell lines that have activating STAT3 mutations with Stattic, a small molecule inhibitor of STAT3 that targets the SH2 domain, resulted in increased apoptosis. This effect was not observed in cell lines without STAT3 mutations [67]. Activating STAT3 mutations are less common in AITL, but were found in four out of 85 cases through a next-generation sequencing approach [62]. In a cohort of 48 patients with ATLL, 25.5% had activating mutations in the SH2 domain of STAT3 [81].

In a cohort of 88 ALK−ALCL patients, 20% had activating mutations in STAT3 and/or JAK1 [22]. In cutaneous ALCL, STAT3 mutations were observed in three out of 29 patients, and JAK1 mutations were seen in six out of 29 patients. Patients with systemic ALCL and mutations in STAT3 and JAK1 also had strong nuclear pSTAT3 staining, although this was not exclusive to patients harboring these mutations, suggesting that there are other methods of constitutive STAT3 activation in ALCL besides activating mutations in STAT3 and JAK1 [22]. A study of cell lines derived from patients with primary and cutaneous ALK− ALCL and BIA-ALCL found that all cell lines grew independently of cytokine signaling. Some, but not all of the cell lines contained activating mutations in JAK1 and/or STAT3. Upon treatment with JAK inhibitors, six out of eight cell lines demonstrated decreased proliferation. This response was independent of JAK1/STAT3 mutation status, but correlated with pSTAT3 expression [60]. These studies suggest that even in cases without a detectable activating mutation in JAK1 or STAT3, dysregulated STAT3 signaling is involved in oncogenesis, and targeting the JAK/STAT pathway can be an effective strategy. Using Stat3^−/−^ mouse embryonic fibroblasts transduced with viruses carrying recurrent mutations in JAK1 and STAT3, it was shown that while mutated JAK1 is sufficient to drive constitutive pSTAT3 expression, mutations in both JAK1 and STAT3 synergize to drive more robust pSTAT3 expression [22]. Treatment of ALK− ALCL tumor bearing mice carrying JAK1 and STAT3 mutations with ruxolitinib, a JAK1/JAK2 inhibitor, led to inhibition of tumor formation, which demonstrates the therapeutic potential of targeting the JAK1/STAT3 pathway [22].

Upstream activation of Janus kinases is seen in other T cell lymphomas besides ALK− ALCL. In a study of 91 patients with mature and immature T cell lymphomas, 31.8% were found to have activating mutations in the JAK/STAT3 pathway, with 13% having JAK3 mutations and 7.7% with JAK1 mutations. Of the patients with mature T cell lymphomas, 18% had mutations in STAT3 [46]. In PBMCs isolated from a patient with SS, STAT3 was constitutively phosphorylated on tyrosine (Tyr) 705, and this was dependent on constitutive tyrosine phosphorylation of JAK3. Treatment with a JAK inhibitor, tyrophostin AG490, inhibited JAK3 and STAT3 activation, and decreased transcription of a STAT3 target gene, IL2RA [42]. Together, these studies show that intrinsic activating mutations in the JAK/STAT signaling pathway are a common feature across different types of T cell lymphomas and that these mutations represent therapeutically relevant and druggable targets.

### 3.2. Activating Kinase Mutations

Constitutive activation of other kinases besides JAK1 is involved in the dysregulation of STAT3, namely NPM-ALK in ALK+ ALCL, as discussed above. NPM-ALK positive cell lines, cell lines transfected with NPM-ALK, and samples from patients with ALK+ ALCL all have STAT3 phosphorylated on Tyr705 and Ser727 [47]. Furthermore, in cell lines, STAT3 phosphorylation leads to increased expression of STAT3 target gene BCL2 like 1 (BCL2L1), and this expression was abrogated by treatment with a dominant negative form of STAT3 [47]. These findings were replicated in vivo in a mouse model in which human NPM-ALK was overexpressed in T cells, which also leads to increased expression of pSTAT3 [49]. STAT3 is required for NPM-ALK mediated lymphomagenesis in mouse embryonic fibroblasts, and loss of STAT3 led to increased apoptosis and decreased tumor formation. In mice injected subcutaneously with an NPM-ALK positive tumor cell line, treatment with an anti-sense oligonucleotide specific for STAT3 led to increased involution and necrosis of tumors and decreased tumor burden. Histological examination revealed increased apoptosis in the tumors [50]. These studies both provide evidence for the essential role of STAT3 in NPM-ALK mediated lymphomagenesis and demonstrate that targeting STAT3 is a viable treatment strategy in vivo.

### 3.3. Positive and Negative Regulators of STAT3

Dysregulation of JAK/STAT signaling is also mediated by disruptions in positive and negative regulators of STAT3. In ALK+ ALCL, in addition to NPM-ALK driven STAT3 activation, malignant cells also express protein phosphatase 2A (PP2A). While the exact mechanism is unknown, it is thought that PP2A plays a role in dephosphorylating STAT3 on serine/threonine residues and stabilizing phosphorylation on Tyr705, which is necessary for STAT3 to bind DNA [25]. Immunoprecipitation revealed that PP2A associates with STAT3, and treatment of ALK+ ALCL cell lines with the PP2A inhibitor calyculin A resulted in decreased pSTAT3, with no impact on total STAT3 [25]. Similarly, treatment of an MF cell line with calyculin A also led to decreased Tyr705 STAT3 phosphorylation, thus preventing STAT3 from binding DNA [72].

An inhibitor of STAT3, protein inhibitor of activated STAT3 (PIAS3), binds directly and specifically to STAT3 and mRNA transcripts for this protein were absent in three out of four ALK+ ALCL lines and in two out of three ALK− ALCL lines [25]. ALK+ ALCL lines also lacked protein expression of PIAS3 [25]. Another negative regulator of STAT3 is Src homology region 2 domain-containing phosphatase-1 (SHP-1), a tyrosine phosphatase that was absent in ALCL. Analysis of CTCL and ALK+ ALCL cell lines revealed decreased SHP-1 mRNA and protein expression. There were no mutations identified within the SHP-1 gene, but the promoter was heavily methylated, which led to silencing of SHP-1. Expression of SHP-1 was correlated with a decrease in pJAK3, which suggests that SHP-1 silencing is another factor that contributes to dysregulated JAK/STAT signaling [51]. Consistent with this, in 44 ALK+ and ALK− ALCL tumor samples, 82% were SHP-1 negative by immunohistochemistry, and this was frequently as a result of methylation of the SHP-1 promoter [52]. When SHP-1 plasmids were transfected into two ALK+ ALCL SHP-1-negative cell lines, pJAK3 and pSTAT3 were decreased, as were downstream targets of STAT3 including Bcl-2 [53]. Expression of SHP-1 led to decreased total protein levels of JAK3 and NPM-ALK, and immunoprecipitation revealed that SHP-1 associates with both of these proteins. While expression of SHP-1 did not lead to increased apoptosis, it did lead to increased G_1_ cell cycle arrest [53]. Collectively, these studies show that STAT3 dysregulation is not only driven by activating mutations in STAT3 and in kinases, but it is also mediated by the loss of negative regulators.

In NKTCL, an additional negative regulator of STAT3, receptor-type tyrosine-protein phosphatase k (PTPRK), demonstrated decreased protein and mRNA expression in NKTCL cell lines and primary samples compared to expression in healthy NK cells. Analysis of five NKTCL cells lines showed that 80% had decreased PTPRK expression, as did 15 out of 27 primary tumors. Similar to SHP-1, the PTPRK propter was hypermethylated, which led to the decreased mRNA and protein expression. Protein expression of PTPRK was inversely correlated with pSTAT3 expression, and immunoprecipitation showed that STAT3 is a direct target of PTPRK. Transduction of NKTCL cell lines with PTPRK caused a decrease in pSTAT3, which suggests that targeting this phosphatase may have therapeutic benefit for these patients [68].

### 3.4. miRNA Regulation of STAT3

Members of the non-coding RNA families, particularly microRNAs (miRNAs), play crucial roles in regulating gene expression. Studies of cells from ATLL patients showed low levels of miR-124a, and restoration of miR-124a expression in a mouse model of ATLL led to decreased tumor formation and growth. [81]. Luciferase assays revealed that *STAT3* is a direct target of miR-124a, and treatment of HTLV-1/ATLL cell lines with a STAT3 inhibitor, S3I-201, led to decreased proliferation. Additionally, a study of changes in gene expression upon expression of miR-124a in ATLL mice revealed decreased expression of known STAT3 target genes such as *survivin* and *SOCS3*, with the greatest decrease seen in *PIM1*, a kinase which has previously been implicated in other leukemias but is not known to be involved in T cell lymphomas. Overexpression of *PIM1* correlates with overexpression of STAT3 in ATLL patient samples, and treatment of ATLL cell lines with a Pim1 inhibitor, AZD1208, led to decreased proliferation, while treatment of ATLL mice with AZD1208 led to decreased tumor formation [81]. The interaction between *PIM1* and STAT3 and potential benefit of combined inhibition in ATLL deserves further investigation.

Another miRNA, miR-155, is overexpressed in ALK− ALCL patient tumors as compared to ALK+ ALCL tumors or normal CD3+ T cells. Transfection of pre-miR-155 into ALCL cell lines with low basal levels of miR-155 led to inhibition of *SOCS1* expression and increased pSTAT3. ALK− ALCL tumors cells treated with anti-miR-155 displayed inhibition of apoptosis, suggesting the presence of an active miR-155-SOCS1-STAT3 axis [61]. These results highlight the presence of non-coding RNA alterations that can directly and indirectly affect normal regulation of STAT3 signaling in T cell lymphomas. The role of other miRNAs and of long non-coding RNAs such as miR-30 and CASC2, which can regulate STAT3 signaling in solid malignancies, needs to be further elucidated in the context of hematological neoplasms [15].

### 3.5. Disruptions in the Tumor Microenvironment

Components of the tumor microenvironment also play a role in maintaining STAT3 activation in T cell lymphomas. In SS, IL-21 is a target of STAT3 that also drives its constitutive expression, forming an autocrine positive feedback loop that is essential for the maintenance of STAT3 signaling. While constitutive STAT3 signaling is often lost when SS cells are cultured, treatment of malignant CD4+ T cells with IL-21 led to robust pSTAT3 expression. Furthermore, stimulation of malignant cells with IL-21 led to increased IL21 mRNA levels, and this was blocked by pre-treatment with a STAT3 inhibitor, illustrating that IL21 is a target of STAT3 and also promotes STAT3 activation [73].

Given that STAT3 activation is such a prevalent feature in mature T cell lymphomas and that there are multiple mechanisms driving its phosphorylation and activation of downstream targets, these studies provide a plethora of potential therapeutic targets.

## 4. Targets of STAT3

### 4.1. BCL-2 Family Members

Phosphorylation of STAT3 through various mechanisms leads to the transcription of several genes involved in cell transformation, expansion, and regulation of the tumor microenvironment [82]. Among the STAT3 target genes that are relevant for malignant transformation are those encoding protein products belonging to the B cell lymphoma 2 (Bcl-2) family of proteins that regulate apoptotic pathways. Inhibiting the DNA binding ability of STAT3 through JAK kinase inhibition has been linked with decreased anti-apoptotic BCL2 expression and increased pro-apoptotic BCL2 Associated X (BAX) expression, along with induction of apoptosis in MF tumor cells [42]. Another Bcl-2 family member, B cell lymphoma extra-large (Bcl-_X_L), has decreased protein expression in the SS patient derived cell line Hut78 upon siRNA knockdown of STAT3, which leads to increased apoptosis [74]. In cells derived from patients with T-LGL, treatment with a JAK kinase inhibitor led to decreased myeloid cell leukemia 1 (Mcl-1) protein expression and increased apoptosis in the majority of samples. This response was independent of Bcl-2 and Bcl-xL, suggesting possible differences in the contribution of these Bcl-2 family member proteins to malignant cell survival across various T cell lymphomas. Treatment with an antisense oligo against STAT3 resulted in decreased STAT3 protein levels, decreased Mcl-1 protein levels, and increased sensitivity to Fas-mediated apoptosis in T-LGL cells [66]. Blunting the activity of STAT3 using a transcription-targeting drug, THZ1, a covalent inhibitor of CDK7, has also been associated with decreased transcription of MCL1, BCL2, BCL2L1), and increased levels of BAX in PTCL [63]. Additionally, in ALK+ ALCL there is a strong association between pSTAT3 positivity and nuclear expression of Survivin, another inhibitor of apoptosis [54]. These results point to a pivotal role of STAT3 in maintaining apoptosis resistance in malignant clones through expression of Bcl-2 family members.

### 4.2. Micro-RNAs

Micro-RNAs play an important role in lymphomagenesis and mir-21, a miRNA previously linked with modulation of PI3K signaling, is also a target of STAT3 [75,83,84]. In SS cells, chromatin immunoprecipitation revealed that miR-21 is a direct target of STAT3. Primary SS cells overexpress miR-21, which is dependent on IL-21/STAT3 signaling [75]. Interestingly, the use of anti-miR-21 oligos in the SS cell line Se-Ax led to increased susceptibility to apoptosis, independent of any effects on downstream PI3K/AKT signaling, suggesting a yet to be elucidated mechanism of miR-21 in promoting malignant cell fitness [75]. A tumor suppressor microRNA, miR-22, is inhibited in CTCL cell lines by STAT5B, STAT3, and JAK3. Treatment with a histone deacetylase (HDAC) inhibitor increased miR-22 expression [76]. In ALK+ ALCL cells, overactive STAT3 signaling induces methylation of miR-29a, a member of the miR-29 family that is capable of modulating Mcl-1 expression [55]. Repression of miR-29 leads to increased levels of Mcl-1 in ALK+ ALCL tumors and decreased apoptosis, possibly accounting for the inferior prognostic outlook seen in ALCL cases with pSTAT3 positive tumors [55].

### 4.3. PD-1/PD-L1 Axis

Programmed cell death protein 1 (PD-1)/programmed cell death-ligand 1 (PD-L1) axis and inhibition of checkpoint signaling in the context of host anti-tumor response is an area of intense investigation, given the therapeutic potential of PD-L1 blockade in a variety of malignancies. In ALCL, STAT3 has been implicated in promoting PD-L1 expression. While fluorescence in situ hybridization analysis did not reveal any amplification of PD-L1/PD-L2 in ALK− or ALK+ ALCL cell lines, PD-L1 protein was highly expressed in several of these lines as assessed by Western blot and immunohistochemistry [56]. In tumor samples, 18 out of 24 ALK− ALCL samples had PD-L1 expression, compared to 18 out of 36 ALK+ ALCL tumor samples, and in both groups PD-L1 expression was correlated with STAT3 activation [56]. Knockdown of STAT3 using siRNA constructs resulted in decreased PD-L1 protein expression [56]. These findings agree with an earlier study that showed that in ALK+ ALCL, PD-L1 is highly expressed at both the mRNA and protein levels, and this is driven by STAT3 through NPM-ALK [57]. STAT3 binds directly to the PD-L1 promoter, and both siRNA knockdown of STAT3 and inhibition of NPM-ALK led to decreased PD-L1 gene expression [57].

Similarly, in a cohort of 30 NKTCL tumor samples, 77% were positive for pSTAT3 by immunohistochemistry, and 93% were positive for PD-L1. Samples with pSTAT3 positivity had much higher PD-L1 expression, suggesting that STAT3 is an important driver of PD-L1 expression [69]. In NKTCL tumor cell lines with constitutive STAT3 phosphorylation and high levels of PD-L1 expression, treatment with Stattic and a STAT3 specific antisense oligonucleotide led to decreased pSTAT3 and PD-L1 expression. In Ba/F3 cells expressing the p.E616K mutation in STAT3, which is located in the SH2 domain, chromatin immunoprecipitation qPCR demonstrated increased binding of STAT3 to the PD-L1 promoter, illustrating that PD-L1 is a target of STAT3 [69]. PD-L1 blockade has shown efficacy in seven cases of refractory NKTCL, and given that STAT3 drives PD-L1 expression, there may be a benefit in combining these therapeutic strategies [70].

### 4.4. IRF4

STAT3 also binds to and regulates the expression of interferon regulatory factor 4 (IRF4), a transcription factor important for promoting Th17 differentiation [21]. Chromatin immunoprecipitation revealed that STAT3 binds to the regulatory regions of IRF4 [58]. The STAT3 dependent expression of IRF4 appears to be involved in oncogenesis in ALK+ ALCL cells [58]. In ALCL cell lines, shRNA knockdown of IRF4 led to decreased cell survival, which was reversed by ectopic expression of IRF4. Knockdown of STAT3 in cell lines also led to decreased IRF4 mRNA and protein expression. In ALCL cell lines with transduced IRF4, STAT3 knockdown led to decreased apoptosis, suggesting that IRF4 plays a role in the STAT3 mediated growth and survival of malignant cells [58]. In ALCL, IRF4 regulates Myc expression, which leads to increased pro-survival signals [59]. In MF cells, IRF4, STAT3, and STAT5B drive secretion of IL-9, and IL-9 blockade after psoralen + UVA (PUVA) leads to increased apoptosis in vitro [77]. Interestingly, MF patients with complete response to PUVA had significantly reduced IL-9 production, suggesting that this cytokine may play a role in maintaining the malignant milieu [77].

### 4.5. IL-17 Cytokine

A pro-inflammatory cytokine abundantly expressed in CTCL lesions is IL-17. STAT3 plays an important role in driving the Th17 differentiation of naïve CD4+ T cells, and it may also drive increased expression of IL-17 in T cell lymphoma [78]. In mice, expression of a hyperactive mutant of STAT3, STAT3C, selectively in T lymphocytes is sufficient to trigger malignant expansion of T cells in the skin of the animals and augment IL-17 production by the T cells. Consequently, the animals develop a disease highly reminiscent of MF [78]. Secretion of IL-17 by malignant cells isolated from CTCL patients is also dependent on JAK3/STAT3 signaling, and treatment with either a small molecule inhibitor against JAK3 or siRNAs targeting either JAK3 or STAT3 led to abrogation of IL-17 production [79].

### 4.6. SOCS3 Protein

Another target of STAT3 is suppressors of cytokine signaling (SOCS). SOCS proteins are transcribed after cytokine signaling and can bind to the kinase domain of JAK kinases to inhibit downstream phosphorylation of STAT3. SOCS3 is a direct target of STAT3, forming a negative feedback loop that regulates response to cytokine signaling [80]. However, in CTCL cell lines SOCS3 is constitutively active and is dependent on constitutive STAT3 expression [85]. Transfection of CTCL cells with a cDNA encoding a dominant negative STAT3 led to decreased SOCS3 expression and increased sensitivity to IFN-α. This indicates that constitutive SOCS3 may play a role in tumor progression by blocking IFN-α mediated growth inhibition and decreasing overall sensitivity to cytokine signaling [85]. As discussed above, IL-21 is also a target of STAT3, and in SS an autocrine feedback loop forms between STAT3 and IL-21, leading to increased STAT3, IL-21, and IL-21 receptor levels [73]. This, along with the role of SOCS3 in promoting insensitivity of malignant cells to cytokine signaling, demonstrates how STAT3 overexpression can shape the tumor microenvironment.

## 5. Therapeutic Targeting of STAT3

STAT3 is a particularly attractive therapeutic target because it is constitutively active in most human solid and hematological tumors, and at the same time only transiently active in normal tissues [86]. In addition to its well-known role in promoting tumor cell proliferation, survival, and metastasis, recent studies also highlight an important role of STAT3 in regulating mitochondrial function and chromatin accessibility in addition to its conventional role as a signal transducer and transcription factor. These observations, all taken together, provide compelling evidence that STAT3 is among the most promising new targets for cancer therapy [87].

Despite the pivotal role played by STAT3 in malignant transformation and tumorigenesis, and the apparent therapeutic potential for targeted STAT3 inhibition, there has been underwhelming progress made in developing selective inhibitors for clinical practice. Due to the widespread nature of STAT3 signaling throughout mammalian organ systems, targeting this pleiotropic signaling pathway is a daunting challenge.

Mechanistically, there are two strategies for targeting STAT3 activation: (1) Inhibition of activating kinases upstream of STAT3 and (2) direct inhibition of the activity and function of STAT3.

### 5.1. Upstream STAT3 Inhibition

While several growth factors and cytokines have been implicated in the activation of STAT3, most efforts have been focused on targeting JAK and Src kinases. Ruxolitinib is a potent JAK1/2 inhibitor, and the first therapeutic of this class to be approved by the U.S. Food and Drug Administration (FDA) for the treatment of intermediate or high-risk myelofibrosis, based on combined results of the COMFORT-I and COMFORT-II Trials. Since then, several inhibitors targeting one or more members of the JAK family of enzymes have received FDA approval for the treatment of myeloproliferative neoplasms (MPNs) as well as several non-malignant disorders including psoriasis, rheumatoid arthritis, and steroid-refractory acute graft versus host disease (aGVHD)—Study INCB 18424-271 (NCT02953678) [88].

However, the role of JAK inhibition as a therapeutic strategy in T cell lymphomas warrants further investigation. A multi-center phase II study evaluating the efficacy of ruxolitinib in relapsed/refractory PTCL and CTCL observed overall response rates (ORR) of up to 40% with a trend towards higher ORR and more durable responses among patients with JAK/STAT alterations [64]. Ruxolitinib suppresses STAT3 and decreases the viability of Epstein–Barr virus (EBV)-positive T or NK cell lines and PBMCs from patients with chronic active Epstein-Barr virus infection (CAEBV) [71]. Interestingly, JAK1/2 inhibitor treatment was associated with an increased risk of progression to aggressive lymphomas in patients with myelofibrosis who have a preexisting B cell clone in their bone marrow [89]. The mechanism remains unknown, but may be related to immunosuppressive properties of JAK1/2 inhibitors on T cell function, and thus immune surveillance, or be due to inhibition of clonal competition and eventual expansion of the malignant B cell clone.

### 5.2. Direct STAT3 Inhibition

Inhibition of upstream kinases may not be the optimal strategy due to multiple upstream activators that converge on STAT3 signaling [87] and the occurrence of activating STAT3 mutations that allow receptor independent homodimerization or enhancement of STAT3 activation in response to ligand binding [90,91,92]. An alternative strategy is direct STAT3 inhibition by targeting one of its three functional motifs: The N–terminal domain, the SH2 domain, or the DNA binding domain (DBD). Broadly, these inhibitors can be categorized into three groups: peptides, small molecules, and oligonucleotides. While numerous preclinical studies on cell lines and animal models justify further development, the use of direct STAT3 inhibitors in clinical trials for cancer therapy is limited by lack of cell penetrance, rapid degradation, lack of specific binding, and concerns for unpredicted toxicities [93].

Peptide therapeutics can be designed to bind specifically and potently to any STAT3 domain but have a poor pharmacokinetic profile due to rapid degradation, instability, and poor membrane penetration [90,91]. In contrast, non-peptide small molecules can cross the cell membrane efficiently and are relatively stable. Multiple small molecule inhibitors targeting the STAT3-SH2 domain have been identified through virtual screening and constitute the largest class of direct STAT3 inhibitors. Most molecules in this class remain in preclinical development due to lack of crucial evidence to demonstrate specific STAT3 binding. STAT3 lacks a classic druggable binding pocket [92], and the high homology among STAT proteins makes specific targeting very difficult [94]. OPB-51602 and OPB-31121 have completed testing in phase I trials for the treatment of advanced malignancies. Of these, only OPB-51602 demonstrated promising antitumor activity, but the toxicities were significant, particularly with continuous administration [93,95]. STA-21 has completed phase I/II trial for psoriasis and showed improvement in psoriatic lesions after 2 weeks of treatment [96]. WP1220 is another small molecule inhibitor initially developed for treatment of psoriasis, currently being tested in a phase II clinical trial for the topical treatment of CTCL.

Lastly, STAT3 decoy oligonucleotides demonstrated activity in lymphoma in preclinical studies, but suffered from poor bioavailability and short half-lives in vivo due to rapid nuclease degradation, which limits their clinical application [96].

Although no direct STAT3 inhibitor has yet received FDA approval for clinical use, considerable progress has been made in preclinical studies. Accumulating evidence continues to support the idea that STAT3 is an important oncogene, critical for the progression of a number of T cell malignant diseases, justifying further efforts to develop therapeutic modalities specifically targeting this cytokine signaling pathway.

## 6. Conclusions

While STAT3 is an essential transcription factor orchestrating various homeostatic processes, including the differentiation of T cells to combat extracellular pathogens, dysregulation of the STAT3 signaling pathway is common in several T and NK cell lymphomas. Work in animal models has demonstrated that aberrant STAT3 signaling is a driver of CTCL and ALK+ ALCL lymphomagenesis. Studies of patient samples have demonstrated that dysregulated STAT3 signaling is a common feature in mature T and NK cell lymphomas and thus represents a promising therapeutic target. This dysregulation can happen through a spectrum of mechanisms such as recurrent somatic mutations, oncogenic kinases, or the inability to modulate phosphorylation, which all converge on the ability to continuously initiate JAK/STAT3 signaling. STAT3 dysregulation not only allows the malignant T cells to out-compete their healthy counterparts, but it also gives them the ability to shape the inflammatory milieu to support further oncogenic signals. In the more severe cases, malignant progression depletes the healthy repertoire of T cells capable of battling intra- and extracellular pathogens, leaving the host at severe risk of life-threatening complications from infections. Achieving a more complete picture of the spectrum of events that lead to dysregulated STAT3 signaling in T cell malignancies and the precise molecular consequences of the augmented STAT3 signal may enable development of future therapeutic modalities for STAT3 driven cancers. 

## Figures and Tables

**Figure 1 cancers-11-01711-f001:**
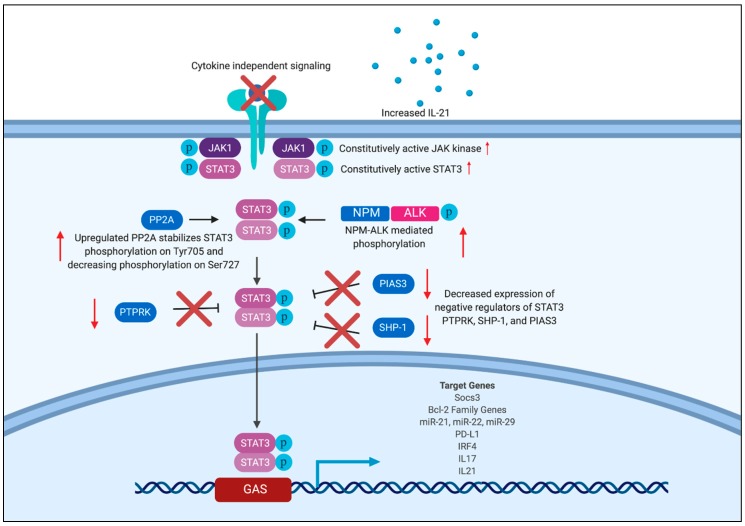
Mechanisms of STAT3 Dysregulation in mature T and NK cell lymphomas and target genes. Mechanisms of STAT3 dysregulation across mature T and NK cell lymphomas include constitutive activation of JAK kinases and STAT3, PP2A stabilization of pSTAT3 on Tyr707, NPM-ALK mediated phosphorylation of STAT3, and decreased expression of negative regulators of STAT3 PTPRK, SHP-1, and PIAS3. While constitutive STAT3 activation in T cell lymphomas is often cytokine independent, in SS IL-21 in the tumor microenvironment is necessary for STAT3 activation. Once in the nucleus, pSTAT3 drives expression of genes important for growth, proliferation, and survival of malignant cells. Janus kinase (JAK), protein phosphatase 2A (PP2A), receptor-type tyrosine-protein phosphatase k (PTPRK) protein inhibitor of activated STAT3 (PIAS3), Nucleophosphomin (NPM), Src homology region 2 domain-containing phosphatase-1 (SHP-1), Sézary Syndrome (SS).

**Table 1 cancers-11-01711-t001:** Mature T and NK cell lymphomas with aberrant STAT3 signaling.

Mature T and NK Cell Neoplasms	Relevant Literature
Anaplastic large-cell lymphoma, ALK+	[22,24,25,26,46,47,48,49,50,51,52,53,54,55,56,57,58,59]
Anaplastic large-cell lymphoma, ALK−	[22,25,46,52,56,59,60,61]
Breast Implant Associated anaplastic large-cell lymphoma	[27,28,60]
Angioimmunoblastic T cell lymphoma	[22,46,62]
Peripheral T cell lymphoma, NOS	[22,46,63,64]
T cell large granular Lymphocytic Leukemia	[27,30,31,32,46,65,66]
NK/T cell lymphoma, nasal type and NK/T cell lymphoma	[27,34,36,46,67,68,69,70,71]
Hepatosplenic T cell lymphoma	[27,33,35]
Primary cutaneous γδ T cell lymphomas	[34]
Peripheral γδ T cell lymphomas	[34]
Enteropathy Associated T cell lymphoma	[37]
Cutaneous T cell lymphoma (Mycosis fungoides and Sézary syndrome)	[41,42,43,44,45,46,51,64,72,73,74,75,76,77,78,79,80]
Adult T cell leukemia/lymphoma	[29,81]

Signal transducer and activator of transcription (STAT), Natural killer (NK), Not otherwise specified (NOS).

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
