# Peer review of "STAT3 Dysregulation in Mature T and NK Cell Lymphomas"

_cancers, 2019, doi:10.3390/cancers11111711_

Round 1
Reviewer 1 Report
The Manuscript ID: cancers-619066n is a well written paper by Angelina Seffens, Alberto Herrera, Cosmin Tegla, Terkild B. Bruus, Kenneth B. Hymes, Niels Odum, Larisa J. Geskin, Sergei B. Koralov, which reviewed the recent contributions to understanding of the role of STAT3 in T cell lymphomagenesis, mechanisms driving STAT3 activation in T cell lymphomas, and current efforts at targeting STAT3 signalling in T cell malignancies.
Some concerns that should be taken into account:
1.- According to the 2008 WHO classification and the 2017 WHO update, T-cell lymphomas encompasses a complex series of mature T- and natural killer (NK) cell neoplasms excluding precursor T-cell neoplasms arising from immature thymocytes. This fact should be enough, in my opinion, to modify the title (a tentative new title could be: “STAT3 dysregulation in mature T- and natural killer (NK) cell lymphomas”); to be commented into the Abstract (for example, line 14: T cell lymphomas comprise a distinct class of non-Hodgkin’s lymphomas, which encompasses a complex series of mature T- and natural killer (NK) cell neoplasms excluding precursor T-cell neoplasms arising from immature thymocytes); and to be commented into the Introduction section (lines 33-34). Furthermore, a few comments should be included in the Introduction section about the different entities recognized into the 2017 update WHO, which gives special importance to the dysregulation of JAK-STAT pathway due to gene mutations which are common to various agressive and indolent neoplasms (see Matutes E, 2018, The 2017 WHO update on mature T-and natural killer (NK) cell neoplasms. Int J Lab Hem (supplement 1): 97-103).
2.- The authors should leave the background of the subject better established by highlighting the most important revisions in their opinion on the role played by deregulation via JAK-STAT and specifically STAT3 in cancer or in haematological malignancies.
For example:
-Pencik et al. (2016). JAK-STAT in cancers: from cytokines to non-coding genome. Cytokine 87: 26-36, which includes a section about JAK-STAT in hematopoietic malignancies with special reference to STAT3.
-Yu et al (2014) Revisiting STAT3 signaling in cancer. Nature Rev Cancer 14, 736.(which highlights new non-canonical roles for STAT 3 in regulating mitochondrion functions and gene expression through epigenetic mechanisms).
-Vainchenker and Contastinescu (2013). The role of JAK-STAT signalling in haematological malignancies. Oncogene 32, 2601-13.
3.- From my point of view, the review focuses excessively in STAT3 without taking into account other members of the STAT family which share mechanisms of action and targets. It would be convenient to include in the introduction some additional data on the structural and functional characteristics of the different proteins encoded by the main members of the STAT family (especially SAT1, 3 and 5). This task would benefit with some additional figure and table. Furthermore, a recent review about the 2017 WHO update on mature T- and natural killer (NK) cell neoplasms (Matures, 2018, Int J Lab Hem (supplement 1): 97-103) highlights the involvement of JAK-STAT pathway in the development of mature T- and NK cell neoplasms. As Figure 1 depicts STAT3 dysregulation seems to be the main event in Mycosis fungoides and Sezary syndrome (MF/SS) and in ALK negative anaplastic large cell lymphoma (ALK-ALCL) but acts in conjunction with other members of the STAT family in T/NK LGL and extranodal T/NK. However STAT5 plays a main role in T-PLL (T-cell prolymphocytic leukaemia), MEILT (monomorphic epiteliothropic intestinal T-cell lymphoma) and Hepatosplenic T-cell ymphoma.
4.- In the section 3, Mechanism of aberrant STAT3 activation, I have missed some comment on the role played by the deregulation of microRNAs and lncRNAs in controlling the expression of STAT3 (see Pencik et al. 2016. JAK-STAT in cancers: from cytokines to non-coding genome. Cytokine 87: 26-36; see also Bellon et al (2016) Constitutive activation of PIM1 kinase is a therapeutic target for adult T-cell leukemia). I have also missed some comment about the epigenetic inhibition of some SOCS family inhibitor proteins.
5.- In the section 4, on STAT3 targets, the authors not mention PIM1, which seems essential in ATL (Bellon et al (2016) Constitutive activation of PIM1 kinase is a therapeutic target for adult T-cell leukemia)
6.-In the section 5, Therapeutic targeting of STAT3, it would be good to comment the review by Yu et al (2014, Revisiting STAT3 signalling in cancer. Nature Rev Cancer 14, 736), which highlights the importance of STAT3 among the most promising new targets for cancer therapy.
Reviewer 2 Report
Seffens A et al extensively reviewed STAT3 dysregulation in T Cell Lymphomas by introducing the overview of STAT3 dysregulation in T cell lymphomas, mechanism of STAT3 dysregulation and therapeutic targeting of STAT3 dysregulation. The information provided in the manuscript will be very useful for the researchers in the field.
Here are my comments:
The authors used quite a few “lymphomagenesis“ in the manuscript. Please note the presence of STAT3 mutation or pSTAT3 in patient sample is not sufficient to support the comments on the role of STAT3 in lymphomagenesis. Although Stat3 was shown to be required for ALK-mediated lymphomagenesis (Chiarle R, Simmons WJ, Cai H, Dhall G, Zamo A, Raz R, Karras JG, Levy DE, Inghirami G. Stat3 is required for ALK-mediated lymphomagenesis and provides a possible therapeutic target. Nat Med. 2005 Jun;11(6):623-9. Epub 2005 May 15.), the role of STAT3 in lymphomagenesis especially non ALK+ T cell lymphoma is not sufficiently evaluated. Specific references should be provided to support the comments on the role of STAT3 in lymphomagenesis. I would be better if subheadings are used for a long description of a topic. It would be helpful for the readers to find specific information without going through the whole topic. For example, under the topic of “Mechanism of aberrant STAT3 activation”, STAT3 mutation, Jak mutation, loss of inhibitor etc. may be used as subheadings. References should be given if it is available. For examples, at Line 44, actively transported through the nuclear pores using importin α/β and RanGDP complexes, there are quite a few references supporting the comment. For example, Ling Liu, Kevin M. McBride, and Nancy C. Reich STAT3 nuclear import is independent of tyrosine phosphorylation and mediated by importin-α3. PNAS June 7, 2005 102 (23) 8150-8155; Sekimoto, T., Nakajima, K., Tachibana, T., Hirano, T. & Yoneda, Y. (1996) Interferon-gamma-dependent nuclear import of Stat1 is mediated by the GTPase activity of Ran/TC4. J. Biol. Chem. 271, 31017–31020. Readers will appreciate it if an overview of pathogenesis of T cell lymphomas is given in the introduction to show the STAT3 dysregulation is only one of the major events. A related review can be cited here.Author Response
Please see attached

Round 2
Reviewer 1 Report
In the revised version, the authors addressed all the points of critique and provided explanatory and additional data.